# Gaming for the Education of Biology in High Schools

**Marina Lantzouni *** , **Vassilis Poulopoulos** and **Manolis Wallace**

ΓAB LAB, Knowledge and Uncertainty Research Laboratory, University of the Peloponnese, Akadimaikou G. K. Vlachou Str., 22131 Tripolis, Greece; vacilos@uop.gr (V.P.); wallace@uop.gr (M.W.)
* Correspondence: m.lantzouni@go.uop.gr

**Definition:** Game-based learning refers to an educational approach where games (digital or analogue) are used in order to engage students in interactive and immersive experiences designed to teach specific concepts, skills or subjects. Gamification refers to the application of game design elements, such as point systems, rewards, narratives, and competition, to non-game contexts. Game elements, mechanics and structures, when incorporated into the learning process, can enhance student understanding and increase engagement, motivation and retention of educational content. Teaching Biology can present challenges mainly due to the complexity of the subject matter, the different scales of biological organisation, and because it often includes challenging and counterintuitive concepts that may contradict students' preconceived notions. Integrating gaming into the high school Biology curriculum not only tackles the challenges of teaching complex concepts but can also promote student engagement. Customising gaming experiences to Biology intricacies enhances critical thinking and creates a dynamic learning environment tailored to the demands of high school biological education. This entry explores the integration of gaming and gamification in high school Biology education to overcome challenges in sustaining student interest. Additionally, the article highlights the diverse applications of games in education, showcasing their versatility in enriching the educational process. Future research should evaluate specific games, explore design principles, and consider challenges associated with implementation. In conclusion, using games in Biology education promises to enhance engagement, promote active learning, and deepen understanding, contributing to narrowing the gap in biological literacy.

**Keywords:** gamification in biology; board games; serious games; gamified education

## 1. Introduction

Teacher-centred methods of teaching Biology in high schools often face challenges in capturing and sustaining students' interest. Gaming and gamification present an opportunity to address this issue by tapping into the natural affinity that students have for interactive and enjoyable experiences. Incorporating gaming into the high school Biology curriculum offers a transformative approach by addressing the unique challenges of teaching complex biological concepts and promoting student engagement. By tailoring gaming experiences to the intricacies of Biology, educators can enhance student engagement, promote critical thinking, and provide a dynamic learning environment that aligns with the specific demands of biological education in high schools.

The term "gamification" is primarily used to describe the application of game elements and mechanisms in non-game environments, with the ultimate goal of enhancing processes and improving the experience of the involved parties [1]. A second definition of gamification suggests that it is the use of the logic and dynamics of games to solve problems and increase the audience's dependence on them [2]. This practice utilises features that make a game enjoyable to enhance user engagement, such as points, badges, tracks, difficulty levels, leaderboards, etc. [3].

It should be noted that gamification first gained popularity in fields other than education [4], as applications have been developed in various fields, such as entrepreneur-

ship, aiming to improve personnel performance or increase and retain customer loyalty (e.g., point collection systems from purchases, visits, and similar rewards). It is also applied in the health sector as motivation for improving physical fitness (e.g., personal trainer applications, pedometers) and as a means for collecting experimental data by research entities, among many other areas of our daily lives.

Students respond better to interactive learning environments [5–8], and the need to shift from the teacher-centred model to a more learner-centred, collaborative model is becoming evident. Games expose children to learning situations that often involve activities slightly beyond their current skill levels, making them valuable in early childhood and later education [9]. The use of games as a medium for learning is not a new idea, as games, regardless of difficulty or complexity, with or without technology, can aid the learning process. Through games, cause-and-effect relationships can be taught, and what is learned through games tends to be better retained by students due to the interactive nature of the learning experience [10].

As the use of digital (mainly) games by children and adolescents outside the school context continues to increase, researchers, educators, and programmers have developed a particular interest in these games. International literature increasingly refers to the use and effectiveness of games in the educational process, and various applications have been developed to facilitate their use by educators (e.g., Kahoot!, Edmodo, quizizz, wordwall, helpfulgames, learning Apps, and many more). Studies show that gamification can improve the motivation and performance of students in a variety of learning environments [11–14].

However, most modern research tends to overlook how individual student characteristics influence the impact of game elements on their engagement and involvement in learning. Research, nevertheless, indicates that the perception and response to game elements vary among students, suggesting a need for personalised gamification [14].

Biology as a subject lends itself to the creation of educational games, and there are numerous gamification tools that can be applied in its teaching. Gamification in Biology can capture students' interest, not only in urgent remote education but also in face-to-face teaching, especially as in some cases (e.g., in Greece) it has been downgraded to a single-hour class in secondary education, leading to a loss of connection between the teacher, students, and the subject. Gamification tools provide the opportunity for further engagement of students without imposing the burden of "homework".

The characterisation of COVID-19 as a pandemic on 11 March 2020 [15] placed all countries in a state of emergency. Since then, the pandemic has affected, among other things, the operation of educational institutions worldwide, as educators and learners are required to adapt to new challenges. The model primarily utilised is that of "emergency remote learning" (ERL). However, this model can easily devolve into a low-interaction model between educators and students, where knowledge is simply transmitted from the transmitter (educator) to the receiver (learner), downgrading the quality and effectiveness of the educational process.

Gamification can play a significant role in upgrading emergency distance education by providing tools to educators to capture students' interest and actively engage them in learning. The lesson becomes more enjoyable, the students' experience improves, and their commitment may increase [16].

Remote learning emerged out of necessity and caught a significant part of the educational community unprepared. However, integrating technology into the educational process will help upgrade it, and it is certain that many of the methods and techniques employed out of necessity will remain even after the pandemic. The educational process and our understanding of pedagogy are evolving.

The game exposes the child to a learning situation, as it often involves activities that are slightly beyond the skills the child has already acquired. For this reason, games have been adopted early on in preschool and later education [9]. The use of games as a means of learning is not a new idea. Regardless of the level of difficulty or complexity and whether or not technology is involved, games can aid the learning process. Through games, causal

relationships can be taught, and what is learned through games tends to be better retained by students, mainly due to the interactive nature of the learning experience [10].

As the use of digital (mainly) games by children and adolescents outside the school context continues to increase, researchers, educators, and programmers have developed a particular interest in these games. International literature increasingly refers to the use and effectiveness of games in the educational process [17–23].

Extensive academic research (e.g., [6,10,17,19,24–26]) has minimised most of the doubts related to the use of video games as learning tools. However, this does not mean in any way that conclusions can be drawn and generalised about all games, all fields, and all learners based on research on the effectiveness of one game in one learning area for one group of learners [10,27].

Through the exploration of the educational value of engaging with science through computer technology, it has been found that knowledge acquisition is better supported as science becomes accessible, thinking becomes visible, students can learn from each other, and autonomous learning is promoted [18,28].

These games could transform not only the way we learn but also individuals at various levels. In the technical report by Hays [29] ways in which games can be used in education are mentioned:

* Assessment of Existing Cognitive Levels
* Performance Measurement on Various Criteria
* Assistance in Evaluating Educational Approaches and Programs
* Providing Educational Information on Specific Knowledge and Skills
* Aiding in Changing Attitudes/Perceptions
* Serving as Pre-Organisers for Other Forms of Education
* Replacing Other Forms of Teaching to Convey Facts, Teach Skills, and Provide Knowledge
* Serving as Training/Practice Mediums, Problem Solving/Exercise
* Aiding in the Integration and Retention of Knowledge and Skills
* Representing the Dynamics or Abstract Concepts of a Cognitive Object

These diverse applications highlight the versatility of games in the educational domain, demonstrating their potential to enhance learning experiences and outcomes in various ways.

This entry paper delves into the potential of incorporating gaming elements into high school Biology education in order to capture and retain student interest in complex biological concepts. This paper explores how well-designed educational games can address this challenge. By examining the unique strengths of game-based learning and its effectiveness in promoting knowledge acquisition, engagement, and critical thinking skills, this paper aims to provide a compelling argument for integrating gaming into the high school Biology curriculum. We will explore various game types, analyse their suitability for different learning objectives, and discuss practical considerations for teachers seeking to implement game-based learning strategies in their classrooms.

## 2. Concepts

Gaming refers to the incorporation of interactive digital or analogue activities, often in the form of video games or simulations, into the high school Biology curriculum. These activities are designed to engage students in the learning process through challenges, scenarios, and problem-solving exercises related to biological concepts.

Game-based learning refers to an educational approach that utilises games, whether digital or analogue, to engage students in interactive and immersive experiences designed to teach specific concepts, skills, or subjects. It involves incorporating game elements, mechanics, and structures into the learning process to enhance student understanding, motivation, and retention of educational content.

Gamification or gameful design, involves the application of game design elements, such as point systems, rewards, narratives, and competition, to non-game contexts, specifically in the realm of high school Biology education. The goal is to enhance student

motivation, participation, and learning outcomes by introducing game-like features into traditional educational experiences.

## 3. Theoretical Framework

The bibliographic review [30] indicates that there are many different game elements that can be included in the educational process, with many tending to appear repeatedly, while some others appear sporadically. Some elements enjoy widespread acceptance, while others may need additional processing and research, and some are new to the educational process, borrowed from video games [30]. Next, we will present some of the principles applied in the design for the gamification of the educational process.

Objectives: Clear, relatively challenging, and immediately achievable [31].

Challenges and missions: Clear, concise tasks with increasing complexity [32].

Adaptability: Personalised experiences, adjusted difficulty levels, and challenges corresponding to each player's level, with increasing difficulty as players' skills expand [32,33].

Progress: Visible progress up to "mastery", achievable through point systems, progress bars, levels (tracks), coins, or XP (experience points).

Feedback: Immediate feedback or short feedback cycles, immediate rewards instead of long-term rewards/benefits [3,33].

Competition and cooperation: Achieved through badges, scoreboards (ranking), tracks, avatars [32,34]. Badges usually do not affect a student's grade but aim to activate and mobilise students within a healthy competition. Badges are awarded for successes in challenges or even for simple participation, time management, performance [30].

Visible Status: Reputation and social recognition with the help of badges, points, scoreboards and avatars [32].

Accessibility: Locked content that unlocks [34].

Freedom of choice: Multiple paths to success allowing students to choose their own sub-goals/steps within the broader framework of goals [32,34].

Freedom to fail: Low risk from submitting deliverables and the possibility of multiple attempts [33]. Students who do not perform well are not "punished" but have the opportunity to review and resubmit assignments or retake a quiz. This specific feature is one of the most controversial elements that one can apply in a traditional classroom, and its value was not adequately highlighted until recently [30].

Storytelling: Narration or dramatisation with avatars [3,32].

New characters and roles, new challenges (through new avatars) [32].

Time constraint: Achieved with the presence of a countdown clock [31].

The characteristics of games that are ultimately incorporated into the educational process through gamification are commitment, reward and recognition, feedback, and participation in communities. All five functions mentioned are high on Maslow's hierarchy of innate human needs, indicating that gamification can have very positive results when applied to the educational process [35].

## 4. Why Is Biology Difficult to Teach

Our experience of 20 years of teaching Biology has shown that it can present challenges due to the complexity of the subject matter, the need for a balance between theoretical and practical aspects, and the diverse range of topics within the discipline.

Biology is everywhere. From news articles about pandemics to the plants they see growing outside, students are surrounded by biology in action. This makes it even more important to help them bridge the gap between what they learn in class and the real world. By encouraging them to find connections between their studies and everyday experiences, science becomes more engaging and relevant [36].

One challenge in teaching Biology stems from students' existing knowledge or preconceptions about the subject. Students arrive in Biology classrooms with a surprising amount of prior knowledge gleaned from personal experiences or informal learning [36]. While

these preconceptions can sometimes align with scientific principles, they may also conflict with new information students are expected to learn.

This is particularly true for complex topics like cellular respiration and photosynthesis. These concepts have been extensively researched using various methodologies, yet student misconceptions about these and other core biological concepts remain poorly understood [37].

For example, although the strength of the evidence supporting evolution has increased markedly since the discovery of DNA and the advances in molecular biology, the public resistance to accepting evolution seems to grow stronger! [38]. It is crucial to acknowledge that addressing students' misconceptions systematically can be particularly challenging when teaching evolution. This difficulty often arises because student resistance to evolutionary concepts may be framed within a religious context [39]. In everyday conversation, "theory" often refers to a guess or hunch. However, in science, a well-established theory is built on a strong foundation of evidence. Some students (and their families) dismiss the "theory of evolution" as "just a theory" despite mountains of evidence supporting it. This misunderstanding underscores the critical need for clear communication in Biology education. When we understand how scientific theories are developed and tested, we can better distinguish them from personal beliefs [39].

In general, many concepts that lie within Biology are both difficult to teach and understand, as they consist of complex mechanisms and procedures. On the one hand, this implies that simplifying the procedure through gamification can be the panacea, but in parallel, as it is difficult to teach, it is far more difficult to transfer the knowledge in a gamified procedure. By trying to transfer knowledge in a game, one should have in mind that the gamified procedure should not only be fun and entertaining but also able to transfer the required knowledge to students, making it easier for the professor to teach and provide the ability to assess during the procedure.

Apart from the aforementioned, Biology has an excess scale starting from the lowest level (molecular/cellular) to complete ecosystems. As such, there is already a very strict strategy within the school procedures to provide with the ability to cover a large part of the curriculum that makes it even more challenging to transfer parts of it within a game.

Moreover, Biology as a subject is not "static". Organisations, the environment, science and even humans evolve through time. This means that educators should be able to adapt to changes that happen around us. If we add the factor that technology has helped the evolution to happen even faster, this ends up in a procedure of endless and lifelong learning for the educators themselves, which, in consequence, leads to continuous updates on the curricula or the way they are transferred as knowledge in a classroom.

Hands-on laboratory work is crucial for Biology education to reinforce theoretical concepts and develop practical skills. However, providing adequate laboratory facilities, materials, and ensuring safety can be logistically challenging for schools, especially those with limited resources. Biology often involves challenging and counterintuitive concepts that may contradict students' preconceived notions. Addressing and correcting misconceptions requires thoughtful pedagogical strategies and time for individualised attention.

Teaching Biology presents its share of challenges, but it is a critical endeavour. By using innovative teaching methods, continuous professional development, and creating a supportive learning environment, we can overcome these hurdles.

Despite these challenges, effective Biology education is crucial for developing scientific literacy, fostering an appreciation for the natural world, and preparing students for future studies and careers in the life sciences. Innovative teaching methods, continuous professional development, and a supportive learning environment can contribute to overcoming these challenges in the teaching of Biology.

Teaching Biology effectively involves understanding how students learn and addressing their existing ideas about the natural world. Research on students' misconceptions can help educators design better lessons and assessments to promote a deeper understanding of biological concepts.

## 5. Approaches

### 5.1. Games and Biology: The Early Steps

In 1991, Kramer [40] presented the possibility of an interactive Biology lesson using electronic computers in a series of articles in *The American Biology Teacher*. The limited software available at that time ran on MS-DOS, and despite these lessons having a classic linear structure, students achieved more than just flip pages. They could choose from the menu which topics they wanted to study in what order, and, most importantly, they were required to answer a question or conduct a virtual experiment, such as a dissection or virtual cultivation. Critics commented that such an approach was far from the actual execution of an experiment. However, supporters argued that it supported learning much more than merely reading about a specific experiment or seeing diagrams or videos of it. Moreover, this method "made learning more fun! A lesson is more like a video game!" [40]. The seed had been planted.

### 5.2. Video Games

The first research on electronic games began around the early 1970s, coinciding with the appearance of the first video games. However, the initial studies that went beyond possible psychological effects and focused on the educational perspective for children and adolescents emerged more than a decade later [41].

Children from a very young age comfortably play online multiplayer games. The detailed manual for each game can be extensive, up to 300 pages. However, many players can effectively handle the game after minimal hours of practice. What educator would not want their students to absorb educational information so quickly? [42]. What do video games have that makes learning so easy? Gentile and Gentile [43] believe that video games encompass many hallmarks of an ideal learning environment. They have clear goals, offer various difficulty levels, and provide continuous feedback.

Continuous feedback is one of the main factors creating a dynamic and powerful learning environment. If students do not have immediate access to feedback, for example, if they must exit the game environment, its effectiveness as a learning tool decreases. However, special attention is needed in designing a game to strike a balance between the assistance given to players [10] and the likelihood of providing them with all the answers as spoon-fed information.

Several noteworthy examples include AntibioGame® [44], a serious game aiming to improve the training of medical students in antibiotic use in primary care; META!BLAST, a serious game to explore the complexities of structural and metabolic cell Biology [45]; Orbis Pictus Bestialis [46], a serious game on ethology, behaviourism and animal learning for high school students; Electron Chute [47], a combination of video and board game on the energy transformation process during photosynthesis, and these are but a few of the available games in the literature.

### 5.3. Biology Board Games

Several individual efforts have been made to develop board games related to various fields of Biology. Some games are commercially available (e.g., Cytosis, Photosynthesis, Wingspan, Evolution, Into the Forest—Nature's Food Chain Game, Plague Inc.), while others have been created by research teams and educators, which teachers can download from the internet, print, and use in the classroom, such as Bioenergy Farm Game, Farming for Ecosystem Services, Geneticist Trumps, and many more. Following that, a brief overview of board games that have been developed for teaching Biology will be provided.

Taylor and Jackson [48] presented a very interesting board game titled ImmunoScenarios, where players progress on a classic game board shaped like an antibody. In each dice roll, they encounter an immunological scenario: they may encounter a new pathogenic microorganism and need to be directed to the hospital, they may be given the opportunity to receive a vaccine against a pathogen, or they may encounter an antigen they have already faced and demonstrate their "immunity cards" to progress.

Signal is a board game presented by Kaur [49] as a fun and engaging pedagogical approach that can promote a greater understanding of content. In this game, the students work as teams to build a working synaptic connection by drawing cards featuring molecules and proteins involved in neurotransmission and placing the cards onto specific locations on the pre- and post-synaptic neurons illustrated on the game board.

"Discovering the Cell" (Célula Adentro) is an investigative board game (in Portuguese) designed to teach complex Cell and Molecular Biology themes to secondary-level students [50], "Synthesizing Proteins" is a board game aiming to help students construct a protein synthesis model [51], "Carbohydeck" is a board game to teach the stereochemistry of carbohydrates to university students [52] and the "Natural Selection Game" is a board game with spoons, chopsticks and gummy bears to teach evolution in high school and entry-level college courses [53] to name but a few.

### 5.4. Games in the Schoolyard—Floor Games

Several attempts have been recorded to take Biology lessons outside the classroom in the form of games. For example, in 2018, a simple team game with a floorboard was presented, where student pawns throw the dice, advance, and must answer a question related to the lesson content within a few seconds [54]. The game, in addition to questions, includes penalty cards or challenge cards that the entire team must perform. Another board game that can be transformed into a floor game is Race to Displace [55], where students, taking on the role of indigenous or exotic (invasive) plants, compete to gain a larger population. The game unfolds on a board with actions (creating seeds, taking over meadows, etc.) and action cards related to competition for light, nutrients, and space.

### 5.5. Escape Rooms

Escape rooms are becoming a trend in gaming design, as they have emerged as a captivating and effective approach to learning in various subjects. They involve active learning; they encourage active participation, encouraging students to solve puzzles and collaborate with peers to "escape". Most common are the digital escape rooms, either for mobile phones or computers, but there are also cases where actual classrooms have been transformed into themed environments, such as in Brady and Andersen [56], where an escape room to teach advanced genetic analysis to university students. Grande-de-Prado explains the concept and the potential of edu-Escape Rooms in an Encyclopedia entry article [57].

### 5.6. Simulations

Another alternative teaching approach that involves teaching through games and "gamifying" lessons is simulations. For instance, there is a simulation for respiration and photosynthesis presented in the *American Biology Teacher* magazine [58], where students, in groups, use plasticised cards and stickers to represent the processes of photosynthesis and cellular respiration. Another simulation is "Finding Garrett" [59], where students are required to develop a new communication system in groups with the aim of highlighting convergent and divergent evolution.

It is evident that a plethora of games—tabletop, floor, and electronic—with Biology themes have been created. However, there does not seem to be a systematic record of them or a comparative evaluation.

## 6. Conclusions

The effect of Biology games on students has been assessed in several studies and in many ways. Student achievement, in general, is usually measured by tests given upon the completion of the corresponding units where a game was implemented, compared with a control group where no games were used for the same unit. Some studies have revealed controversial findings. For example, [60–62] discuss that didactic games are not more effective in teaching than conventional methods, while [6,57,63,64] (to name but a

few) find the games more effective than the traditional teaching methods. Furthermore, students who played games in Biology lessons have expressed a decrease in their anxiety towards Biology lessons upon completion of the corresponding unit taught [63].

In conclusion, the integration of gaming and gamification into high school Biology education seems to hold tremendous potential to revolutionise the learning experience. Games, when applied to teaching, can be used as creative and interactive methods to capture students' interest and convey the relevance of Biology to their lives. While further research is needed to fully evaluate the effectiveness of specific games across diverse contexts and biological topics, the current body of evidence suggests that incorporating games into the curriculum can be a valuable tool in fostering student engagement, learning and achievement. Future research should continue to explore the design principles that optimise game-based learning for Biology education, investigate the long-term impacts of game use on student learning, and consider the potential challenges and limitations associated with implementation.

The use of games in Biology education holds a significant promise for enhancing student engagement, promoting active learning, and fostering a deeper understanding of biological concepts. By harnessing the power of games, educators can ignite student engagement in biological exploration and plant the seeds of lifelong curiosity about the intricacies of the natural world and the scientific processes that govern it, ultimately playing a crucial role in narrowing the gap in biological literacy.

**Author Contributions:** Conceptualisation, M.L. and V.P.; writing—original draft preparation, M.L.; writing—review and editing, M.L., V.P. and M.W. All authors have read and agreed to the published version of the manuscript.

**Funding:** This research received no external funding.

**Conflicts of Interest:** The authors declare no conflicts of interest.

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
