# Peer review of "Gaming for the Education of Biology in High Schools"

_encyclopedia, doi:10.3390/encyclopedia4020041_

Round 1

Reviewer 1 Report

Comments and Suggestions for Authors  

1.  In the Abstract, it should be "game-based learning" rather than "game based learning". Add a hyphen.

2. I personally suggest making "the traditional methods of teaching" more specific. Because what you mean is "outdated teaching methods", but regardless of how advanced teaching methods are, as long as they have been used for a long time, they can be called "traditional". So the expression "the traditional methods" is not accurate. Terms like audiolingual method, rote learning, teacher-centered lectures, etc., can more accurately convey your intended meaning.

3.  I suggest revising the content of the second paragraph on the second page. You stated, "gamification is not exclusive to the educational process", but in fact, gamification first gained popularity in fields such as entrepreneurship, marketing, and health, and then developed in the field of education. I suggest checking or modifying it.

4.  All citations in your article are incorrect. Citations "(4), (5), (6), (7)" should be "[4-7]". Throughout the text, it should be "[...]". I suggest using reference management software like EndNote or Zotero, using the "IEEE proceeding" format for bibliography management.

5.  Please add references for each definition in the "concepts" section. If necessary, discuss the differences and similarities between different concepts, or define them based on different references.

6.  The "Why is biology difficult to teach" section lacks research methods or corresponding data/resources. I suggest revising it.

7.  I believe disciplines should be capitalized. It should be "Biology" instead of "biology".

8.  Some parts have no citations. Please add some.

Author Response

Thank you very much for taking the time to review this manuscript and provide thoughtful feedback  on our paper. Please find the detailed responses below and the corresponding revisions/corrections highlighted/in track changes in the re-submitted files.

Point-by-point response to Comments and Suggestions for Authors

Comment1. In the Abstract, it should be "game-
based learning" rather than "game based learning". Add a hyphen.

Response1. We appreciate your valuable comment and we replaced the words “game based” with game-based” throughout the text.

Comment2. I personally suggest making "the traditional methods of teaching" more specific. Because what you mean is "outdated teaching methods", but regardless of how advanced teaching methods are, as long as they have been used for a long time, they can be called "traditional". So the expression "the traditional methods" is not accurate. Terms like audiolingual method, rote learning, teacher-centered lectures, etc., can more accurately convey your intended meaning.

Response2. Thank you for your valuable comment. We will use teacher-centred learning

Comment3. I suggest revising the content of the second paragraph on the second page. You stated, "gamification is not exclusive to the educational process", but in fact, gamification first gained popularity in fields such as entrepreneurship, marketing, and health, and the n developed in the field of education. I suggest checking or modifying it.

Response3. We thought our audience would be more “education-focused” so we wanted to highlight the “non exclusiveness’ of gamification to education. We welcome your valuable comment and  we are modifying the text to demonstrate the origin.

Comment4. All citations in your article are incorrect. Citations "(4), (5), (6), (7)" should be "[4- 7]". Throughout the text, it should be "[...]". I suggest using reference management software like EndNote or Zotero, using the "IEEE proceeding" format for bibliography management.

Response4. We used Mendeley for the references. The IEEE format used parentheses instead of brackets, and we didn’t notice it. We downloaded an update and now brackets are ok. But for the consecutive we will have to do by hand, at the end of the revision. Thank you very much for pointing it out.

Comment 5. Please add references for each definition in the "concepts" section. If necessary, discuss the differences and similarities between different concepts, or define them based on different references.

Response5. Thank you for your comment. The definitions of the concepts are discussed throughout the paper, here we thought of summarising, and we didn’t thing that they should be as extended with references, comparisons.

Comment6. The "Why is biology difficult to teach" section lacks research methods or corresponding data/resources. I suggest revising it.

Response6. We appreciate your valuable comment and we revised the section “why is biology difficult to teach”. We have also re-organised the entry, after a comment of Reviewer 2.

Comment7. I believe disciplines should be capitalized. It should be "Biology" instead of "biology".

Response7. We will capitalize Disciplines throughout the text. (Biology instead of biology) 

Comment 8. Some parts have no citations. Please add some.

Response8. Thank you for your comment on the citations. We have added the citations we missed. 

Reviewer 2 Report

Comments and Suggestions for Authors

The entry "Gaming for the education of biology in High Schools" represents a considerable contribution on how game-based learning improve the learning experience of Biology in high school students. However, I suggest a major revision to improve the clarity of the entry. The suggestions are outlined below:

- I suggest reorganising the entry. First, the theoretical framework could appear after the introduction section. Thus, the games reviewed (approaches section), in my opinion, are the main contribution of the entry. I suggest that the concepts section should be incorporated into the theoretical framework.

-I suggest that the authors should apply the content set out in the theoretical framework to the games of Biology reviewed.

-The definitions section is clear and concise, but could be improved. For example, the idea stated in lines 24 to 26 is repeated. I suggest deleting this sentence. I propose that the definition section could be called summary.

-The introduction section clearly does not state the importance of the work. I suggest that the author state at the end of the introduction section, what is the main purpose or contribution of this entry.

Author Response

Thank you very much for taking the time to review this manuscript and provide thoughtful feedback  on our paper. Please find the detailed responses below and the corresponding revisions/corrections highlighted/in track changes in the re-submitted files.

Comment1. The entry "Gaming for the education of biology in High Schools" represents a considerable contribution on how game-based learning improve the learning experience of Biology in high school students. However, I suggest a major revision to improve the clarity of the entry. The suggestions are outlined below:

Response1. Thank you very much for your positive comments, which are encouraging and motivating.

Comment2. I suggest reorganising the entry. First, the theoretical framework could appear after the introduction section. Thus, the games reviewed (approaches section), in my opinion, are the main contribution of the entry. I suggest that the concepts section should be incorporated into the theoretical framework.

Response2. We appreciate your valuable comment and we are re-organising the entry: Theoretical Framework will be n.3, why biology is difficult n.4 and Approaches will be n.5. It makes more sense like that, and thank you for pointing it out!

Comment3. -I suggest that the authors should apply the content set out in the theoretical framework to the games of Biology reviewed.

Response3. Thank you for your comment. We appreciate it and we think that with the reorganisation of the paper it is more clear now.

Comment4. -The definitions section is clear and concise, but could be improved. For example, the idea stated in lines 24 to 26 is repeated. I suggest deleting this sentence. I propose that the definition section could be called summary.

Response4. Thank you for your valuable comment. You are right about repeating the sentence in the “Definition” section. We cannot rename the section to “summary”, as it is an entry paper for the Encyclopedia, and we were instructed to do it this way (Definition).

Comment5. -The introduction section clearly does not state the importance of the work. I suggest that the author state at the end of the introduction section, what is the main purpose or contribution of this entry.

Response5. We take into consideration your valuable comment and therefore we added, at the end of the introduction section, a clear description of the main contribution of this entry.

Round 2

Reviewer 1 Report

Comments and Suggestions for Authors  

On page 3, line 115, there's a space only in the front part, without the back part. Please add it. The specific location is "(e.g., [6, 10, 17, 19, 24-26]". On line 123 of the same page, there should be a space before "[18, 28]". Please add it.

My personal recommendation would be to remove links to several software, specifically the footer content on the second page (Kahoot!, Quizziz, Wordwall, and Helpful Games). This precaution is due to the fact that web pages are susceptible to modification or hacking. In the past, many educational materials included links that led to hacked websites, resulting in adult content or gaming sites being displayed. While these particular software links may not present this issue, I still advise their removal. Just remove the footnotes suffices. The same precaution applies to the seventh page.

I suggest changing the order of the three Concepts to "gaming", "game-based learning", and "gamification". Gaming implies using full-fledged video games, game-based learning implies using full-fledged or half-fledged video games for learning, gamification implies using "totally not full-fledged games" (only game elements) for other purposes. So, gamification should not appear between the other two concepts. Just change the order.

Please add references in the Concepts section. Definitions cannot be your own. Add sources.

I suggest checking the presentation format of DOI in the reference list. Normally it should be "https://doi.org/10XXX". Please avoid the presentation format "doi: XXX". You can input the reference list into Crossref, Crossref will provide accurate DOI addresses and presentation formats. The link is https://doi.crossref.org/simpleTextQuery."

Author Response

We would like to express our gratitude for your valuable feedback. Your comments have greatly improved the clarity and strength of our manuscript. We appreciate the time and effort you dedicated to providing constructive criticism.

Following your advice, we have updated the manuscript as follows:

We added the missing parenthesis, and the missing space on page 3.

We removed the links of the software (ie Kahoot!, Quizziz etc) on the second page, and we have also removed the links of the commercial games (Cytosis, Wingspan, Evolution etc) on the seventh page  as you suggested, as a precaution due to the fact that web pages are susceptible to modification or hacking. (In the past many educational materials included links that led to hacked websites, resulting in adult content being displayed).

We changed the order of the three concepts, to: gaming, game-based learning and gamification.

Finally, we added the https://doi.org/ to the DOI format in all the citations that had a DOI. Thank you for pointing out the crossref tool, as it helped us find some missing DOIs!

Thank you again for your valuable comments.

Reviewer 2 Report

Comments and Suggestions for Authors

The manuscript has improved significantly. Therefore, I consider accepting it in its current form.

Author Response

We would like to express our gratitude to you for your valuable feedback. Your comments have greatly improved the clarity and strength of our manuscript. We appreciate the time and effort you dedicated to providing constructive criticism. 

Round 3

Reviewer 1 Report

Comments and Suggestions for Authors

In certain parts of the text, there is an extra line space between paragraphs. Please check and ensure consistency. On page 5, "[35]" should come before the period rather than after. Please try to avoid this exact error. On page 7, line 360, there are periods both before and after "[56]". Please make sure to delete the one before it. Within the main text, "and" should be used between two authors instead of "&". That is, in the article, it should appear as "Gentile and Gentile" on page 6. Please check. Please confirm whether the journal names in the references should be abbreviated or in full. They should be in full. On page 8, in the "funding" section, there is a missing period. I suggest using authors' family names instead of abbreviations in the Author contributions section, such as "M.L.". The second reference contains repeated authors and duplicated publication dates. Please check. The 16th reference is not in English. I personally recommend reviewing its necessity; otherwise, remove it from the manuscript.

Author Response

Thank you for your comments.

We've deleted the extra lines between paragraphs and corrected the periods (missing and extras). We've replaced Gentile & Gentile with Gentile and Gentile.

In the authors' contributions section we've used abreviations following the journal's template. 

We corrected the repeated authors and duplicated publication dates in Reference [2]. Reference [16] is very important for our work so we cannot remove it. 

We appreciate you taking the time to provide serious and constructive feedback. Your comments strengthen our work.